# Towards Compact and Certified Robust DNNs Against Semantic Perturbations

## Abstract

Compactness and robustness are both critical for deploying DNN models, yet most prior work focuses on optimizing one aspect. Few efforts work on obtaining compact DNNs that maintain consistent predictions under semantic mutations, such as changes in facial expression or illumination. To fill this gap, we propose **C**ompression-**A**ware Semantic **R**obustness (CAR) training scheme. Inspired by prior studies on model loss landscapes, we design a composite training objective that guides the pruning mask optimization toward flatter loss regions. We further explicitly incorporate certification conditions on semantically mutated data and enforce consistency between the soft mask used during training and the hard binary mask deployed at inference. The pruned models obtained via CAR consistently achieve higher robustness than the baselines, with improvements of 17%–64% on CelebA-HQ and Flowers-102 across ResNet-18, GoogLeNet, and MobileNet-V2, while maintaining task accuracy comparable to the corresponding no-prune models.

## 1 Introduction

Deep neural networks (DNNs) have demonstrated superior performance across many downstream tasks (Kirillov et al., 2023; Ravi et al., 2024; Vaswani, 2017; Achiam et al., 2023). Owing to extensive research on adversarial samples (Goodfellow et al., 2015; Madry et al., 2018), it is now well recognized that task accuracy alone is not a sufficient measure of model prediction, it is also important to ensure consistent and correct predictions, i.e., robustness, when the input undergoes transformations that do not alter its semantics. Early studies on model robustness primarily targeted pixel-level perturbations, i.e., adding $L_p$-norm bounded noise (Sehwag et al., 2020; Gui et al., 2019; Cohen et al., 2019; Jia et al., 2020), whereas more recent research (Yuan et al., 2023; Mirman et al., 2021) has shifted toward semantic perturbations, e.g., facial expression changes, or illumination variations that more closely reflect real-world conditions.

However, another hurdle for real-world deployment is the prevalence of resource-constrained settings, where model compression techniques such as pruning are commonly applied (Han et al., 2015). This raises concerns that reducing model size could very likely degrade model robustness as well. In a nutshell, our goal is to have *an effective approach for obtaining compact yet robust DNNs that can withstand semantic perturbations*. The most recent related effort to jointly address model compression and robustness is HYDRA (Sehwag et al., 2020), which optimizes the pruning mask with a robust training objective. However, it only accounts for pixel-level perturbations, and our empirical tests show that its pruned model performs poorly against realistic semantic perturbations, as illustrated in Figure 1(c). Another classic strategy to improve model robustness, particularly against semantic transformations, is to leverage emerging generative models (Preechakul et al., 2022; Kim et al., 2022) to synthesize data with targeted semantic variations and incorporate them during pruned model training and fine-tuning. This approach does provide some benefits, but it remains unsatisfactory, particularly at higher compression ratios, since robustness cannot be preserved even with extra data augmentation, as denoted by "LMP" in Figure 1(c). Prior efforts (Hao et al., 2022; Mirman et al., 2021; Yuan et al., 2023) to advance semantic certification methods without involving model compression, are orthogonal to our work, and we provide detailed discussion in Section 5.

To address this research gap, we formulate a joint optimization framework that explicitly incorporates robustness requirements into the compression (prune mask) training process, similar to prior work (Sehwag et al., 2020). However, unlike HYDRA (Sehwag et al., 2020), whose training loss

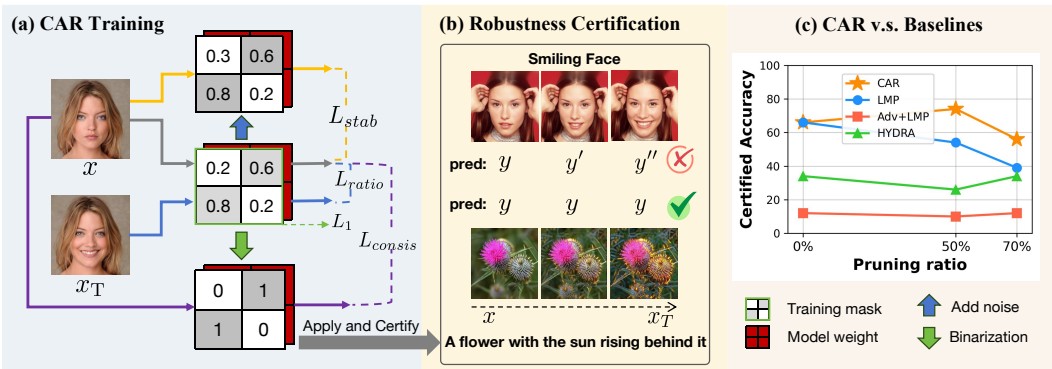

Figure 1: Overview of compression-aware semantic robustness. (a) Training process of our proposed CAR framework, where pruning masks are optimized with stability under mask perturbations $\mathcal{L}_{\text{stab}}$, robustness under semantic input variations $\mathcal{L}_{\text{ratio}}$, soft–hard mask consistency $\mathcal{L}_{\text{consis}}$, and $L_1$ regulation, instead of relying on heuristic magnitude-based pruning. (b) Examples of semantic transformations include varying the intensity of smiles in facial images and altering sunlight levels in flower images, which are used to evaluate the robustness of the deployed model (w/ binary masks). (c) Performance comparison across various pruning ratios shows that our proposed method, CAR, consistently achieves higher robustness than the baselines.

targets pixel-level perturbations, we revisit the fundamental definition of robustness and approach it from the loss landscape perspective, drawing inspiration from related studies (Li et al., 2018). Specifically, we propose a **C**ompression-**A**ware Semantic **R**obustness training scheme, CAR. Central to our approach is a composite training loss design, as illustrated in Figure 1(a). Prior theoretical and empirical studies (Foret et al., 2020; Mi et al., 2025; Li et al., 2024) indicate that the geometry of the loss landscape, particularly the flatness of its minima, is closely linked to model generalization and robustness. Motivated by this, we introduce a stability loss $\mathcal{L}_{\text{stab}}$ that explicitly reduces prediction variance across stochastic compression operators as shown in Figure 1(a), thereby encouraging a flatter loss landscape for model finetuning, as demonstrated in Figure 2. Additionally, we design a margin ratio loss $\mathcal{L}_{\text{ratio}}$ that effectively leverages synthetic data generated by diffusion models to explicitly enhance model robustness against semantic perturbations. Finally, we address the gap between the soft masks optimized during training and the hard binary masks used at deployment by enforcing a soft-to-hard mask consistency loss $\mathcal{L}_{\text{consis}}$, ensuring that the learned robustness transfer effectively to deployable binary masks.

**Key contributions**: We make the following key contributions.

• We propose a compression-aware robustness training scheme to obtain compact yet robust DNNs capable of withstanding semantic perturbations. By revisiting the fundamental definition of model robustness and designing a composite training loss, our approach effectively guides optimization toward flatter minima, identifying a subnetwork, i.e., a prune mask with a targeted compression ratio, that achieves significantly better robustness against semantic perturbations compared to baselines, as shown in Figure 1(c).

• We evaluate the proposed approach on three lightweight DNN architectures, ResNet-18, MobileNet-v2, and GoogleNet, using CelebA-HQ and Oxford Flowers-102 datasets. Notably, CAR produces compressed models at a 50% compression ratio, achieving the highest certified robustness accuracy of 74% on CelebA-HQ and 88% on Flowers-102 using 1k augmented samples, improving by 17%-64% over the baselines and even surpassing the no-prune model by 8% and 4%, while maintaining task accuracy comparable to the no-prune model. Our ablation study further validates the effectiveness of each design loss in contributing to the overall performance.

## 2 CAR: COMPRESSION-AWARE SEMANTIC ROBUSTNESS

We propose CAR, an training scheme that explicitly incorporate the impact of model compression on model robustness and effectively guide model optimization toward not only globally minimal but also smoother regions of the loss landscape, yielding lightweight and robust models.

## 2.1 PROBLEM SETTING

Given a classification task represented by the dataset $\mathcal{S} = (x_i, y_i)_{i=1}^m$ (where $x_i \in \mathbb{R}^n$ and $y_i \in \{1, \ldots, K\}$), we consider a DNN-based classifier $f : \mathbb{R}^n \rightarrow [0,1]^K$ that outputs a probability distribution over $K$ classes for each input. The final prediction is given by the class with the highest probability, i.e., $y_{\text{pred}} = \arg\max_k f(x)_k$.

In terms of model robustness, we adopt a probabilistic certification framework (Pautov et al., 2022), as deterministic counterparts (Cheng et al., 2017; Tjeng et al., 2019) often fail to effectively handle large-scale DNNs and semantic perturbations in real-world scenarios.

**Definition (Probabilistic Robustness).** For an input $x$ with true class $c$ and transform $T : \mathbb{R}^n \rightarrow \mathbb{R}^n$, we obtain the set of transformed inputs $\mathbb{S}_T = \{x_T\}, x_T = T(x)$. The model $f$ is said to be probabilistically robust at $x$ with confidence $1 - \varepsilon$ if the probability of its predictions on $\mathbb{S}_T$ satisfies the following condition:

$$\mathbb{P}_{x_T \sim \mathbb{S}_T}\Big( \arg\max_k f(x_T)[k] = c \Big) \geq 1 - \varepsilon \tag{1}$$

Building on the above definition of robustness, we quantify the prediction discrepancy between $x$ and its transformed counterpart $x_T = T(x)$ as

$$Z(x; T) \triangleq ||\mathbf{p}(x) - \mathbf{p}(x_T)||_\infty \tag{2}$$

where $\mathbf{p}(x) = f(x)$ denotes the class-probability vector with entries sorted in descending order, $p_1 > p_2 > \cdots > p_K$. We measure the margin between the top-2 predictions of input $x$ as $d(x) \triangleq \frac{p_1 - p_2}{2}$. Thus, a sufficient condition for satisfying probabilistic robustness defined in Equation 1 is $Z(x; T) < d(x)$, and we provide the complete proof in Appendix A.2. To summarize, optimizing probabilistic robustness is equivalent to increasing the certified passing probability

$$\max \mathbb{P}\big(Z(x; T) < d(x)\big) \tag{3}$$

which can be achieved by minimizing the prediction discrepancy $Z$ or by maximizing the margin $d$.

In our work, we primarily adopt semantic mutation as the targeted transformation $T$, as it is more realistic and practical in real-world scenarios compared to pixel-based manipulation.

**Definition (Semantic Mutation).** Suppose $x \in \mathbb{R}^n$ corresponds to a latent representation $z \in \mathbb{R}^d$ with $d \ll n$. A semantic-level mutation shifts this latent point linearly in a specified direction to $z'$, from which a new semantically altered input $x'$ is generated through a generative model $G$: $x' = G\big(z + ||\delta|| \cdot \mathbf{s}\big)$. Here, $0 \leq ||\delta|| \leq 1$ specifies the mutation extent bounded by 1, while $\mathbf{s}$ denotes a unit vector indicating the mutation direction. For example, as illustrated in Figure 1, if $x$ is a human face, $\mathbf{s}$ may represent the latent direction toward a smile, with varying $||\delta||$ values corresponding to different smile intensities. Alternatively, if $x$ is a flower, $\mathbf{s}$ may represent the latent direction toward increased sunlight on the flower. The latent-space perspective enables a controllable, single-factor semantic transformation $T$, which we subsequently integrate with our training scheme.

For model compression, we primarily employ unstructured pruning (Han et al., 2015).

**Definition (Model Pruning).** For the DNN classifier $f$, we denote its compressed version as $f_C(\cdot) = f(\cdot; m_C \odot \theta)$, where $m_C$ is the compression mask. In the case of a hard mask, $m_C \in \{0, 1\}$, with 0 indicating a pruned parameter and 1 indicating a preserved parameter. The prune ratio $\text{pr} = 1 - ||m_C||_0 / |\theta|$ reflects the model size reduction, where $||m_C||_0$ denotes the number of ones.

## 2.2 OUR APPROACH

Our objective is to obtain a compressed DNN that not only meets the sparsity constraints but also enhances semantic certified robustness. To this end, we formulate a joint optimization framework that explicitly incorporates certification conditions into the compression training process.

**Compression as a Learnable Mask Parameter.** Instead of rigidly applying magnitude-based pruning (Han et al., 2015), CAR samples compression operator from a probability distribution $C \sim \mathcal{Q}_\phi$ and treats the corresponding mask $m_C \in [0, 1]$ as a learnable coefficient that partially scales the model parameters, subject to the constraint $||m_C||_0 \leq k \cdot |\theta|$, where $k$ denotes the target

compression level. In this way, model compression and robustness optimization could be jointly addressed rather than decoupled, leading to improved outcomes.

**Robustness Optimization via Tightening the Upper Bound of $Z$.** As shown in Equation 3, minimizing the prediction discrepancy $Z$ improves model robustness. For a compression instance $C^\star$, the upper bound of $Z$ can be derived via triangle decomposition as follows:

$$
\begin{aligned}
Z_{C^\star}(x; T) &= \left\| \mathbf{p}_{C^\star}(x) - \mathbf{p}_{C^\star}(x_T) \right\|_\infty \\
&= \left\| \mathbf{p}_{C^\star}(x) - \bar{\mathbf{p}}(x) + \bar{\mathbf{p}}(x) - \bar{\mathbf{p}}(x_T) + \bar{\mathbf{p}}(x_T) - \mathbf{p}_{C^\star}(x_T) \right\|_\infty \\
&\leq \left\| \mathbf{p}_{C^\star}(x) - \bar{\mathbf{p}}(x) \right\|_\infty + \left\| \bar{\mathbf{p}}(x) - \bar{\mathbf{p}}(x_T) \right\|_\infty + \left\| \bar{\mathbf{p}}(x_T) - \mathbf{p}_{C^\star}(x_T) \right\|_\infty \\
&\leq \underbrace{\sqrt{\left\| \mathbf{p}_{C^\star}(x) - \bar{\mathbf{p}}(x) \right\|_2^2}}_{(A)} + \underbrace{\left\| \bar{\mathbf{p}}(x) - \bar{\mathbf{p}}(x_T) \right\|_\infty}_{(B)} + \underbrace{\sqrt{\left\| \bar{\mathbf{p}}(x_T) - \mathbf{p}_{C^\star}(x_T) \right\|_2^2}}_{(C)},
\end{aligned} \tag{4}
$$

where $\bar{\mathbf{p}}(x) = \mathbb{E}_C[\mathbf{p}_C(x)]$ denotes the *ensemble mean* prediction over compression instances. If our training scheme is designed to tighten the above upper bound, we can effectively promote model robustness.

*(i) Minimizing Term (A) and (C) via Stability Loss.* Given two compression instances $C_m, C_n \overset{\text{i.i.d.}}{\sim} \mathcal{Q}_\phi$, we propose a stability loss as follows.

$$
\begin{aligned}
\mathcal{L}_{\text{stab}} &= \mathbb{E}_x \, \mathbb{E}_{C_m, C_n} \left[ \| \mathbf{p}_{C_m}(x) - \mathbf{p}_{C_n}(x) \|_2^2 \right] \\
&= 2 \, \mathbb{E}_x \, \mathbb{E}_C \left[ \| \mathbf{p}_C(x) - \bar{\mathbf{p}}(x) \|_2^2 \right]
\end{aligned} \tag{5}
$$

The full proof is provided in Appendix A.3. Comparing Equation (5) and Equation (4), it is noted that minimizing $\mathcal{L}_{\text{stab}}$ reduces the compression bias at $x$ and $x_T$, i.e., terms (A) and (C), thereby helping tighten the upper bound on $Z$. In essence, $\mathcal{L}_{\text{stab}}$ penalizes prediction variance across compression, guiding the model toward smoother loss minima that typically yield greater robustness.

*(ii) Recursive Minimization of Term (B).* Term (B) represents the difference induced by semantic mutation, averaged across compression instances. It is isomorphic to $Z$ expression. Therefore, once the upper bound of $Z$ is tightened by minimizing Terms (A) and (C), Term (B) is implicitly minimized as well, creating a positive feedback loop that drives all terms toward convergence at minimal values.

**Robustness Optimization via Minimizing Margin-aware Ratio.** Building on Equation (3), we define the normalized robustness ratio

$$
r(x; T) \triangleq \frac{Z(x; T)}{d(x) + \epsilon} \tag{6}
$$

where a small $\epsilon > 0$ is included to avoid division by zero. We then minimize a margin-normalized loss hinged on $\max(r(x; T) - \eta, 0)$, $\eta \in (0, 1]$, where $\eta \in (0, 1]$ is a safety factor, to encourage the certification condition in Equation (3). We further employ softplus as a smooth surrogate for the hinge function $\max(\cdot, 0)$, enabling stable gradient-based optimization.

$$
\mathcal{L}_{\text{ratio}} \triangleq \mathbb{E}_{x, T} \left[ \text{softplus} \left( \max \left( r(x; T) - \eta, 0 \right) \right) \right] \tag{7}
$$

This loss penalizes margin violations ($r > \eta$), guiding the optimization toward reducing $Z(x; T)$ or enlarging $d(x)$.

**Enforcing Soft-Hard Mask Consistency.** While CAR uses soft masks $m_C$ to facilitate optimization, the final deployment requires hard binary masks. To ensure that model performance remains consistent after binarization, we integrate both soft and hard masks into the training process. Specifically, for each batch we compute soft-mask predictions under two independent samples $C_m, C_n \sim \mathcal{Q}\phi$ to evaluate $\mathcal{L}_{\text{stab}}$, apply a semantic transform $x_T$ to compute $\mathcal{L}_{\text{ratio}}$, and simultaneously evaluate a hard-thresholded mask $C^\star$ to measure its divergence from the soft predictions. This discrepancy is penalized via an additional Kullback–Leibler (KL) divergence loss as follows:

$$
\mathcal{L}_{\text{consis}} = \mathbb{E}_x \left[ \text{KL} \left( \mathbf{p}_{C_m}(x) \, \| \, \mathbf{p}_{C^\star}(x) \right) \right] \tag{8}
$$

Gradients are backpropagated through the non-differentiable threshold using a straight-through estimator (Bengio et al., 2013), allowing all forward passes to contribute jointly within each batch.

---

**Algorithm 1** Compression-Aware Semantic Robustness

---

**Input:** DNN ($\theta$), augmented dataset $\mathcal{S}$, pruning ratio pr, initial percentile $\tau$.
**Output:** Compressed network $\theta_f$.
1: Pretrain DNN: $\theta_{\text{pre}} = \arg\min_\theta \mathbb{E}_{(x,y)\sim\mathcal{S}}[L_{\text{ce}}(\theta, x, y)]$
2: Initialize soft mask $m_C$ for each layer: $m_C^i = \frac{\theta_i}{Q(|\theta_i|, \tau)}$
3: Minimize compression-aware robustness loss: $m_C' = \arg\min_{m_C} \mathbb{E}_{(x,y)\sim\mathcal{S}}[L_{\text{CAR}}(\theta_{\text{pre}}, m_C, x, y)]$
4: Mask binarization: $\hat{m} = \mathbf{1}(|m_C'| > |m_C'|_k), |m_C'|_k = k$-th descending percentile of $|m_C'|$, $k = 100 - \text{pr}$
5: Finetune the pruned network: $\theta_f = \arg\min_\theta \mathbb{E}_{(x,y)\sim\mathcal{S}}[L_{\text{ce}}(\theta_{\text{pre}} \odot \hat{m}, x, y)]$

---

This formulation ensures that the stability and margin properties learned under $m_C$ are faithfully transferred to the hard masks required at inference.

**Putting it All Together.** In addition to the above-defined losses, we further incorporate an $L_1$ regularization term on the soft mask $m_C$ to promote model compression. The overall loss function is thus given by:

$$\mathcal{L}_{\text{CAR}} = \lambda_{\text{stab}}\,\mathcal{L}_{\text{stab}} + \lambda_{\text{ratio}}\,\mathcal{L}_{\text{ratio}} + \lambda_{\text{consis}}\mathcal{L}_{\text{consis}} + \lambda_{\text{L}_1}\,\mathcal{L}_{\text{L}_1} \tag{9}$$

The overall training procedure of CAR is summarized in Algorithm 1, consisting of three main stages. First, we train the full-precision model using the conventional cross-entropy loss $\mathcal{L}_{\text{ce}}$. Next, we search for an optimized pruning solution guided by our proposed total loss $\mathcal{L}_{\text{CAR}}$ in Equation (9). Finally, we binarize the resulting soft mask from the previous step and continue fine-tuning the model with the cross-entropy loss $\mathcal{L}_{\text{ce}}$. Inspired by prior works (Sehwag et al., 2020; He et al., 2015) and our empirical study, we adopt partially scaled initialization (Line 2 in Algorithm 1), where the soft mask values in each layer are set proportional to the pretrained parameters and scaled by a coefficient. Specifically, each soft mask value is initialized relative to a percentile of the parameter magnitudes, expressed as $m_C^i = \frac{\theta_i}{Q(|\theta_i|, \tau)}$, where $\tau$ denotes the chosen percentile. For instance, when $\tau = 10\%$, $Q(|\theta_i|, \tau)$ returns the value at the top 10-th percentile in the $i$-th layer. In this way, 10% of the mask values will set to 1, while the remaining entries take proportional values within $[0, 1)$.

## 3 EVALUATION SETUP AND METHODOLOGY

This section presents our experimental setup and evaluation methodology.

**Datasets with Augmented Semantic Transformations**. We evaluate our approach on two classification datasets augmented with semantic transformations: (1) CelebA-HQ (Na et al., 2022), a facial identity dataset containing 307 identities with 4,263 training images and 1,215 testing images. We then employ DIFF-AE (Preechakul et al., 2022) to transform each face into a "smiling" version with varying degrees of intensity. (2) Oxford Flowers-102 (Nilsback & Zisserman, 2008), which includes 102 flower categories with a total of 7,370 images, split into 6,552 for training and 818 for testing. We then employ DiffusionCLIP (Kim et al., 2022), a text-guided diffusion model, to generate mutated images of flowers toward the prompt "A flower in focus, with the sun rising behind it, casting a warm golden glow" at varying intensities. We augment $q$ randomly selected training images and include them directly in the training set, with $q = 1000$ as the default unless stated otherwise. We set $\|\delta\| = 1$ for both diffusion models, producing $x_T$ with the largest semantic perturbation.

**Evaluation Models and Training Settings**. We adopt ResNet-18 (He et al., 2016), GoogLeNet (Szegedy et al., 2015), and MobileNet-V2 (Sandler et al., 2018) as the evaluation model. The initial pretraining stage is conducted for 50 epochs with a learning rate of 0.01. The compression-aware robustness optimization runs for 100 epochs. Finally, the model is fine-tuned for 50 epochs with a learning rate of 0.001. All experiments are implemented in PyTorch and executed on a single NVIDIA A6000 GPU.

**Comparison Baselines**. To the best of our knowledge, no prior work has jointly addressed model pruning and semantic-level certified robustness. Therefore, we draw baselines from the closest related efforts in pixel-level robustness and adopt the following as comparison baselines: (1) Vanilla Train (no prune): the model is trained using cross-entropy loss, with augmented data for training. (2) AdvTrain (no prune) (Madry et al., 2018): the model is trained with both cross-entropy loss and adversarial loss,

using augmented data for training. (3) LMP (Least Magnitude Pruning) (Han et al., 2015): follows train–prune(one shot)–finetune process with cross-entropy loss, incorporating augmented data during both training and finetuning. (4) AdvTrain+LMP: combines the train–prune(one shot)–finetune process with cross-entropy loss and adversarial loss, with augmented data in both training and fine-tuning. (5) HYDRA (Sehwag et al., 2020): follows a train–prune–finetune pipeline, where the pruning mask is learned using the adversarial loss from TRADES (Zhang et al., 2019).

**Evaluation Metrics**. We evaluate two main metrics: (1) task accuracy, i.e., classification accuracy on the test dataset, and (2) probabilistically certified accuracy (PCA), which measures robustness. For certification, we adopt the CC-Cert framework (Pautov et al., 2022). Specifically, we sample 100 test samples, set $\varepsilon = 10^{-3}$ for Equation (1), and compute PCA as follows:

$$\text{PCA}(\mathcal{S}, \varepsilon) = \frac{1}{m} \sum_{i=1}^{m} \mathbf{1} \left[ \mathbb{P}_{x_T \sim \mathbb{S}_T(x_i)} \left( \arg \max_k f(x_T)[k] = y_i \right) \geq 1 - \varepsilon \right] \qquad (10)$$

Here, $\mathcal{S} = (x_i, y_i)_{i=1}^{m}$ denotes the dataset, $m = |\mathcal{S}|$ is the size of the dataset, $f(\cdot)$ is DNN classifier. PCA of $80\%$ with $\varepsilon = 10^{-3}$ means that for $80\%$ of the test samples, the estimated probability of prediction change under the considered semantic transformations is bounded by $10^{-3}$. Further details of the certification framework are provided in Appendix A.4.

## 4 EMPIRICAL RESULTS

We address the following research questions through our evaluation:

**RQ1**: Can CAR produce compressed models with superior accuracy and robustness?

**RQ2**: Do the design components of CAR effectively contribute to the final performance?

**RQ3**: How is CAR's performance influenced by various factors and hyperparameter settings?

### 4.1 OVERALL PERFORMANCE

**Comparison against baselines.** Table 1 presents the performance of our approach and baselines on ResNet-18. For the adversarial loss used in AdvTrain, we employ an $L_2$-norm PGD attack, setting the perturbation budget to $\epsilon = 0.5$ for CelebA and $\epsilon = 2.0$ for Flowers. It is observed that both AdvTrain and HYDRA exhibit lower accuracy and robustness, even compared to LMP. This suggests that strategies effective for pixel-level perturbations do not directly translate to improved robustness against semantic-level mutations, and could even degrade robustness. However, our approach achieves an accuracy of 82.30% and the highest PCA of 74% with 1,000 augmented data samples, even surpassing the vanilla no-prune model by 8% in terms of PCA.

**Performance@Augmentation Intensity.** As shown in Table 1, incorporating more augmented data during training and fine-tuning generally improves robustness, i.e., PCA. However, this is not always guaranteed. For the Flowers dataset, LMP with 4k augmented samples performs significantly worse than with 1k samples. A potential reason is that while additional augmented data increases diversity, it may also introduce noise and distribution shifts. As a result, LMP fails to maintain focus on semantic features and loses consistency under perturbations, leading to reduced PCA. In contrast, our approach can benefit from additional augmented data, demonstrating greater robustness capacity.

**Loss Landscape Visualization.** To illustrate why CAR achieves superior performance in Table 1, we visualize the loss landscapes (Li et al., 2018) of models trained with the conventional adversarial loss versus our proposed loss in Equation (9). As shown in Figure 2(a), the model trained with adversarial loss exhibits a sharp valley around its minimum, indicating that small parameter perturbations can cause a significant increase in loss, thereby reducing robustness. In contrast, our approach in Figure 2(b) yields a much flatter and smoother loss landscape. Prior studies have shown that flatter minima are correlated with improved robustness against perturbations (Foret et al., 2020; Mi et al., 2025; Li et al., 2024), which aligns with the higher PCA observed in our experiments. These results suggest that CAR not only compresses the model but also steers optimization toward more stable regions in the parameter space, thereby enhancing certified robustness.

**Performance across Various DNNs.** Building on the results in Table 1, we select the best-performing baselines and further evaluate them on additional DNN architectures, with results summarized in

| Benchmark / Method | #Aug Data | CelebA-HQ Accuracy | CelebA-HQ PCA | Flowers102 Accuracy | Flowers102 PCA |
|---|---|---|---|---|---|
| **Vanilla Train (no prune)** | 0 | 83.62% | 50% | 97.80% | 58% |
| | 1,000 | **82.96%** | 66% | **97.56%** | 84% |
| | 4,000 | **83.79%** | 71% | **96.70%** | 81% |
| **AdvTrain (no prune)** | 0 | 72.26% | 2% | 90.33% | 28% |
| | 1,000 | 72.59% | 12% | 88.14% | 32% |
| | 4,000 | 72.02% | 14% | 86.80% | 26% |
| **LMP (pr=50%)** | 0 | 83.87% | 46% | 88.51% | 71% |
| | 1,000 | 82.47% | 54% | 89.24% | 71% |
| | 4,000 | 81.98% | 66% | 87.90% | 55% |
| **LMP (pr=70%)** | 0 | 62.06% | 20% | 40.22% | 11% |
| | 1,000 | 63.70% | 39% | 70.90% | 46% |
| | 4,000 | 67.90% | 46% | 69.93% | 36% |
| **AdvTrain+LMP (pr=50%)** | 0 | 73.09% | 5% | 88.51% | 25% |
| | 1,000 | 71.19% | 10% | 89.24% | 27% |
| | 4,000 | 69.22% | 39% | 85.21% | 23% |
| **AdvTrain+LMP (pr=70%)** | 0 | 71.69% | 4% | 89.36% | 28% |
| | 1,000 | 67.65% | 12% | 86.92% | 25% |
| | 4,000 | 70.95% | 22% | 85.45% | 26% |
| **HYDRA (pr=50%)** | 0 | 53.33% | 28% | 79.46% | 51% |
| | 1,000 | 51.44% | 26% | 77.87% | 63% |
| | 4,000 | 51.19% | 48% | 77.87% | 64% |
| **HYDRA (pr=70%)** | 0 | 51.19% | 28% | 79.10% | 56% |
| | 1,000 | 48.97% | 34% | 76.16% | 64% |
| | 4,000 | 53.99% | 40% | 78.61% | 68% |
| **CAR (pr=50%)** | 1,000 | 82.30% | **74%** | 96.45% | **88%** |
| | 4,000 | 79.09% | **74%** | 96.33% | **91%** |
| **CAR (pr=70%)** | 1,000 | 76.13% | 56% | 94.62% | 79% |
| | 4,000 | 76.46% | 63% | **96.70%** | 90% |

Table 1: Comparison of task accuracy and PCA (certified robustness) across our approach and baselines shows that CAR consistently achieves the superior trade-off on both evaluation datasets.

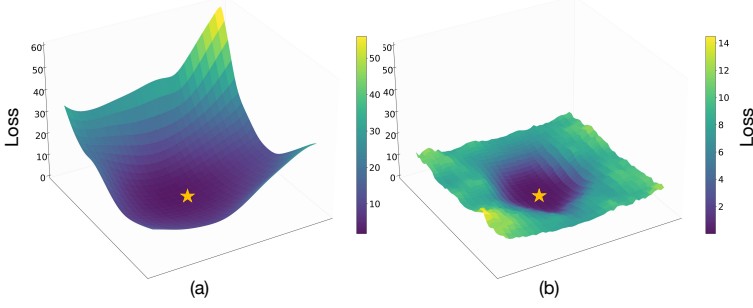

(a)   (b)

Figure 2: Loss landscapes of models trained with (a) adversarial loss and (b) CAR loss in Equation (9). Orange stars mark the location of the initial model. XY plane represents perturbations along two directions in parameter space, while Z axis denotes the cross-entropy loss.

Table 2. Our approach consistently achieves the highest certified robustness across all DNNs, even surpassing the vanilla no-prune model by 8% to 27%. Meanwhile, task accuracy remains comparable to that of the vanilla no-prune model. Among the pruned models, the most striking result appears on GoogleNet, where the PCA gap reaches 42% (49 vs. 7).

| Method / Model | Vanilla Train (no prune) | | LMP (pr=50%) | | CAR (pr=50%) | |
|---|---|---|---|---|---|---|
| | Accuracy | PCA | Accuracy | PCA | Accuracy | PCA |
| MobileNet-v2 | **71.85%** | 33% | 33.91% | 6% | 67.98% | **44%** |
| GoogLeNet | 67.16% | 22% | 44.12% | 7% | **70.86%** | **49%** |
| ResNet-18 | **82.96%** | 66% | 82.47% | 54% | 79.09% | **74%** |

Table 2: Task accuracy and PCA performance across different DNNs on CelebA dataset.

| Metric / Case | CAR | w/o $\mathcal{L}_{\text{stab}}$ | w/o $\mathcal{L}_{\text{ratio}}$ | w/o $\mathcal{L}_{\text{consis}}$ | w/ MSE $\mathcal{L}_{\text{consis}}$ | w/o $\mathcal{L}_{\text{L}_1}$ |
|---|---|---|---|---|---|---|
| Accuracy | 82.72% | 82.72% | 81.89% | 83.37% | 82.14% | 83.05% |
| PCA | **74%** | 68% ($\downarrow 6\%$) | 66% ($\downarrow 8\%$) | 67% ($\downarrow 7\%$) | 65% ($\downarrow 9\%$) | 71% ($\downarrow 3\%$) |

Table 3: Ablation Study: Evaluation under the absence of stability loss $\mathcal{L}_{\text{stab}}$, ratio loss $\mathcal{L}_{\text{ratio}}$, regularization loss $\mathcal{L}_{\text{L}_1}$, and consistency loss $\mathcal{L}_{\text{consis}}$, as well as its MSE-based variant.

## 4.2 Evaluation of Design Effectiveness and Hyperparameter Impact

**Ablation Study for Design Effectiveness.** To evaluate the contribution of each individual loss component to the final performance, we conduct ablation experiments by excluding them one at a time, including stability loss $\mathcal{L}_{\text{stab}}$, margin ratio loss $\mathcal{L}_{\text{ratio}}$, regularization loss $\mathcal{L}_{\text{L}_1}$, and consistency loss $\mathcal{L}_{\text{consis}}$, as well as testing an alternative consistency loss using MSE in place of KL in Equation (8). The experimental results are shown in Table 3. Removing each individual term leads to a notable degradation in certified robustness, highlighting both the effectiveness and necessity of our design. Moreover, the KL-based consistency loss outperforms its mean squared error (MSE) variant, with PCA improving from 65% to 74%.

**Impact of Initialization.** In our empirical study, we observe that the initialization of $m_C$ impacts the optimization process and the final results. We compare three initialization strategies: (1) Random: each $m_C$ value is randomly initialized within $[0, 1]$; (2) Full-scaled (Sehwag et al., 2020): each $m_C$ value is heuristically assigned proportional to its corresponding model parameter value; (3) Partial-scaled (ours): as designed in Algorithm 1, with hyperparameter $\tau$, evaluated over $\tau \in 10\%, 20\%, 30\%, 40\%, 50\%$. Note that full-scaled initialization is a special case of partially scaled initialization with $\tau = 0\%$.

Table 4 shows that models initialized with full-scaled or partial-scaled with $\tau = 50\%$ fail to produce feasible results. In contrast, partial-scaled initialization with $\tau = 30\%$ achieves the best trade-off. This highlights that extreme initialization strategies are suboptimal for finding a robust pruned network. The results show that both strictly adhering to the initial parameter magnitudes ($\tau = 0\%$) and preserving too many connections initially ($\tau = 50\%$) lead to failed outcomes. We further visualize the parameter distributions of the resulting models from Vanilla Train, LMP, and CAR ($\tau = 30\%$). Figure 3 illustrates the distributions for both fully connected and convolutional layers. It can be observed that LMP retains almost no parameters near zero region, whereas CAR preserves an appropriate proportion. Relating this to the results in Table 1, preserving some small-magnitude parameters seems help maintain balanced accuracy, efficiency, and certified robustness.

| Metric / Method | Random | $\tau = 50\%$ | $\tau = 40\%$ | $\tau = 30\%$ | $\tau = 20\%$ | $\tau = 10\%$ | $\tau = 0\%$ (Full-scaled) |
|---|---|---|---|---|---|---|---|
| Accuracy | 78.11% | 0.49% | 80.82% | 82.72% | 82.55% | **83.05%** | 0.49% |
| PCA | 59% | 0% | 63% | **74%** | 66% | 71% | 0% |

Table 4: Impact of initialization methods with pr=50%.

**Impact of Training Hyperparameters.** In addition to initialization, the training process of CAR involves some hyperparameters. We evaluate two key ones: (1) $\delta$: To implement the stability loss $\mathcal{L}_{\text{stab}}$ in Equation (5), we explicitly inject random noise into $m_C$ during the forward pass,

$m'_C = (m_C + \epsilon), \epsilon \sim \mathcal{U}(-\delta, \delta)$, where $\delta$ controls the noise magnitude and $m'_C$ is clipped to the range $[0, 1]$. (2) $\eta$: the safety factor in the margin ratio loss $\mathcal{L}_{\text{ratio}}$ in Equation (7). Table 5 reports the sensitivity of both task accuracy and robustness performance to different settings of the above training hyperparameters. We observe that a moderate noise level of $\delta = 0.5$ yields the highest PCA (74%), suggesting that introducing appropriate stochasticity enhances robustness, while excessive noise (e.g., $\delta = 0.8$) undermines stability. In addition, setting $\eta = 1.0$ achieves the best PCA (74%), whereas using smaller values degrades performance, where the case of $\eta = \infty$ is equivalent to the w/o $\mathcal{L}_{\text{ratio}}$ case in our ablation study (Table 3).

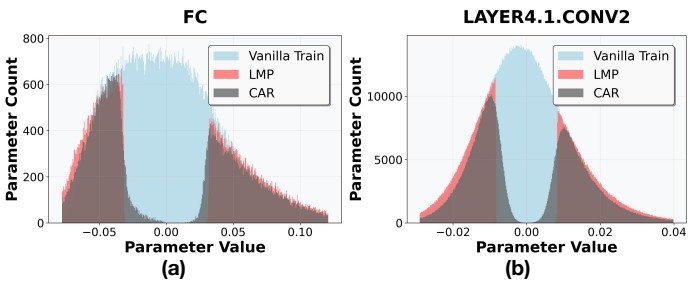

Figure 3: The parameter distributions for fully connected layer and `layer4.conv2`.

| Hyper-Para | $\delta$ (default=0.5) | | | | $\eta$ (default=1) | | | |
| Metric | 0 | 0.1 | 0.5 | 0.8 | $\infty$ | 0.95 | 0.98 | 1.0 |
|---|---|---|---|---|---|---|---|---|
| Accuracy | 82.72% | 82.55% | 82.72% | 82.30% | 81.89% | 82.06% | 82.55% | 82.72% |
| PCA | 68% | 70% | **74%** | 66% | 66% | 67% | 68% | **74%** |

Table 5: Impact of training hyperparameters.

## 5 RELATED WORK

DNN robustness to semantically mutated inputs has recently gained attention, as it better reflects real-world scenarios. Many efforts have been devoted to advancing semantic certification methods, i.e., measuring a model's robustness within a targeted input transformation distribution. In our work, we adopt CC-Cert (Pautov et al., 2022) as the certification method for evaluation, because it imposes no constraints on DNN size and allows the use of off-the-shelf generative models without retraining. In contrast, alternatives such as GenProve (Mirman et al., 2021) are limited to DNNs with nearly 200k parameters, GSmooth (Hao et al., 2022) relies heavily on surrogate models requiring substantial training effort, and GCert (Yuan et al., 2023) is restricted to GANs rather than diffusion models, which have been shown to better generate samples with desired semantic transformations. In terms of the interplay between model compression and robustness, prior works (Gui et al., 2019; Sehwag et al., 2020; Ye et al., 2019; Shumailov et al., 2019; Diffenderfer et al., 2021; Piras et al., 2024) have combined adversarial training with compression techniques (Cheng et al., 2024; He & Xiao, 2024) to obtain compact yet empirically robust models. But they mainly focus on $\ell_p$-bounded pixel-level perturbations, rather than semantic perturbations. In contrast, our work explicitly targets certified semantic robustness in compressed networks and proposes CAR to address this gap.

## 6 CONCLUSION

This work presents CAR, an effective approach for obtaining compact yet robust DNNs that can withstand semantic perturbations through a carefully designed training objective. CAR is comprehensively evaluated to demonstrate its superiority over baselines and its generalization across different DNN architectures. In future work, we aim to extend CAR to handle multi-attribute semantic mutations, develop systematic schemes to mitigate potential biases introduced by generative models, and reduce the overhead of offline generation for probabilistic certification.

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

# A APPENDIX A

## A.1 THE USE OF LARGE LANGUAGE MODELS (LLMS)

We used a large language model (OpenAI ChatGPT, GPT-5) solely for language polishing and grammar checking. All main content, research design, experiments, and conclusions are entirely the work of the authors.

## A.2 PROOF FOR LABEL INVARIANCE.

We denote $c = argmax\ \mathbf{p}$ and $\tilde{c} = argmax\ \mathbf{p}_T$. If $\|\mathbf{p}(x) - \mathbf{p}(x_T)\|_\infty < d = \frac{p_1 - p_2}{2}$ holds, then $\tilde{c} = c$.

*Proof.* Assume, for contradiction, that $\tilde{c} \neq c$. In this case one has

$$p_{T\tilde{c}} > p_{Tc}, \qquad p_c > p_{\tilde{c}}.$$

Moreover, the condition $\|\mathbf{p} - \mathbf{p}_T\|_\infty < d$ implies that

$$|p_k - p_{Tk}| < d \quad \text{for all } k \in \{1, \ldots, K\}.$$

In particular, for indices $c$ and $\tilde{c}$ it follows that

$$p_{Tc} > p_c - d, \qquad p_{T\tilde{c}} < p_{\tilde{c}} + d.$$

Consequently,

$$p_{Tc} - p_{T\tilde{c}} > (p_c - d) - (p_{\tilde{c}} + d) = p_c - p_{\tilde{c}} - 2d.$$

On the other hand, the assumption $p_{T\tilde{c}} > p_{Tc}$ requires $p_{Tc} - p_{T\tilde{c}} < 0$. This contradicts the fact that the right-hand side above is non-negative whenever $p_c - p_{\tilde{c}} \geq 2d$. Hence, the assumption $\tilde{c} \neq c$ is invalid, and we conclude that $\tilde{c} = c$. $\square$

## A.3 VARIANCE IDENTITY.

*Proof.* Let $C_m, C_n \overset{\text{i.i.d.}}{\sim} Q_\phi$ and $\bar{\mathbf{p}}(x) = \mathbb{E}_C[\mathbf{p}_C(x)]$. Then

$$
\begin{aligned}
\mathbb{E}_{C_m,C_n}\left[\|\mathbf{p}_{C_m}(x) - \mathbf{p}_{C_n}(x)\|_2^2\right] &= \mathbb{E}\|\mathbf{p}_{C_m}(x)\|_2^2 + \mathbb{E}\|\mathbf{p}_{C_n}(x)\|_2^2 - 2\,\mathbb{E}\langle\mathbf{p}_{C_m}(x), \mathbf{p}_{C_n}(x)\rangle \\
&= 2\,\mathbb{E}_C\|\mathbf{p}_C(x)\|_2^2 - 2\left\langle\mathbb{E}_{C_m}\mathbf{p}_{C_m}(x),\ \mathbb{E}_{C_n}\mathbf{p}_{C_n}(x)\right\rangle \\
&\quad \text{(by independence of } C_m \text{ and } C_n) \\
&= 2\,\mathbb{E}_C\|\mathbf{p}_C(x)\|_2^2 - 2\,\|\bar{\mathbf{p}}(x)\|_2^2 \\
&= 2\,\mathbb{E}_C\left[\|\mathbf{p}_C(x) - \bar{\mathbf{p}}(x)\|_2^2\right].
\end{aligned}
$$

$\square$

## A.4 CERTIFICATION FRAMEWORK: CC-CERT

According to Equation (2) and Equation (3), and the label invariance condition is $Z < d$. Hence, our goal is to bound $P(Z \geq d)$ from the above. Using Markov's inequality:

$$\mathbb{P}(Z \geq t) \leq \frac{\mathbb{E}(Z)}{t}, \tag{11}$$

where $Z$ is a non-negative random variable and $t \in \mathbb{R}^+$. This provides an initial bound, but CC-Cert refines this using the Chernoff-Cramer inequality (Boucheron et al., 2003):

$$\mathbb{P}(Z \geq d) = \mathbb{P}(e^{Zt} \geq e^{dt}) \leq \frac{\mathbb{E}(e^{Zt})}{e^{dt}}. \tag{12}$$

This gives a tighter upper bound for $\mathbb{P}(Z \geq d)$, which serves as an upper bound for $\varepsilon$ in Equation (1). The optimal value of $t$ is chosen to minimize this bound. However, the true expectation of $e^{Zt}$ is difficult to compute directly.

To address this, CC-Cert (Pautov et al., 2022) estimates $\mathbb{E}(e^{Zt})$ by sampling $n$ transformed inputs $\{x_{T_i}\}_{i=1}^n$, calculating $Z_i = \|\mathbf{p} - \mathbf{p}_{T_i}\|_\infty$ for each, and using:

$$Y = \frac{1}{n \cdot e^{dt}} \sum_{i=1}^n e^{Z_i t}. \tag{13}$$

$Y$ is an estimate of the upper bound on $\varepsilon$. While $Y$ could overestimate or underestimate the true value, underestimation is more problematic as it could lead to false certification. To mitigate this risk, we sample $Y$ independently $l$ times, resulting in $\{Y_1, \ldots, Y_l\}$. The probability of underestimation is then bounded using the following inequality (derived from Pautov et al. (2022); Paley & Zygmund (1930)):

$$\mathbb{P}\left(\frac{\max\{Y_1, \ldots, Y_l\}}{\alpha} < \frac{\mathbb{E}(e^{Xt})}{e^{dt}}\right) < \left(\frac{1}{1 + \frac{n(1-\alpha)^2}{C_v^2}}\right)^l, \tag{14}$$

where $X$ is a random variable in $[0, 1]$, $\alpha$ is a hyperparameter, and $C_v = \frac{\mathrm{Var}(e^{Xt})}{\mathbb{E}(e^{Xt})} \sim 1$ is a coefficient related to $e^{Xt}$.

To obtain a reliable estimate of the upper bound on $\varepsilon$, we repeat the following process $l$ times. For each repetition, we sample $n$ transformed images $\{x_{T_i}\}_{i=1}^n$ from the semantic transformation set $\mathbb{S}_T$ for a given input $x$, compute $Z_i = \|\mathbf{p} - \mathbf{p}_{T_i}\|_\infty$ for each, and calculate the corresponding estimate $Y_i$ using Equation (13). The maximum of these $l$ independent estimates, $\max\{Y_1, \ldots, Y_l\}$, is then used as a conservative upper bound for $\varepsilon$. This ensures that, for sufficiently large $n$ and $l$, the probability of underestimating the true $\varepsilon$ (and therefore falsely certifying a non-robust model) becomes arbitrarily small. This provides a high-confidence probabilistic robustness guarantee.

For the settings, we set $\alpha = 0.9$, $n = 100$, and $l = 10$ in Equation (14), and sweep the temperature vector $t$ logarithmically over $[10^{-4}, 10^4]$ with 500 intervals.

