# OpenReview forum: "Towards Compact and Certified Robust DNNs Against Semantic Perturbations"
_ICLR.cc/2026/Conference — ICLR 2026 Conference Withdrawn Submission_

### Official Review · Reviewer_9sfv · 2025-10-29

**Soundness:** 2
**Presentation:** 2
**Contribution:** 3
**Rating:** 4
**Confidence:** 3

**Summary:**

This paper proposes a compression-aware training scheme to obtain compact DNNs that are robust against semantic perturbations, improving certified robustness while maintaining accuracy with the corresponding no-prune models.

**Strengths:**

1. Proposes specialized losses to explicitly enhance semantic robustness in compressed DNNs.

2. Evaluation across three models and two datasets demonstrates the effectiveness of the losses.

3. Ablation studies validate the contribution of each proposed loss component to the overall performance.

**Weaknesses:**

1. The approach primarily integrates semantic perturbation-aware losses into an existing pruning&robustness framework (HYDRA). While this represents a valuable contribution to the semantic robustness domain, the core training scheme offers limited novelty beyond this adaptation.
2. Comparisons are made against outdated baselines (5 years old). Recent $\ell_p$-bounded pixel-level robustness methods could be adapted by substituting their perturbations with semantic ones.
3. Evaluation on small-scale datasets raises concerns about scalability. How does CAR scale to CIFAR or larger datasets?
4. The four loss components require balancing, but the paper lacks a discussion of these trade-offs.

**Questions:**

1. How does CAR's prediction accuracy under semantic or $\ell_p$ perturbations? The current evaluation of prediction accuracy appears to focus solely on unperturbed test images.

2. If adapted for $\ell_p$ robust pruning methods by replacing $X_T$ with $\ell_p$ adversarial examples, could CAR outperform $\ell_p$-robust pruning methods under comparable experimental settings?

3. Why do most baseline methods exhibit performance degradation after introducing augmented data?

---

### Official Review · Reviewer_zXLc · 2025-10-30

**Soundness:** 2
**Presentation:** 2
**Contribution:** 1
**Rating:** 2
**Confidence:** 4

**Summary:**

This paper introduces a method, Compression-Aware Semantic Robustness, which can be used to produce sparse models that are certifiably robust to semantic perturbations. The CAR loss function involves balancing several terms, including stability, a margin ratio, and hard mask consistency. The method uses this loss to identify a non-binary mask, binarize the mask, and prune the resulting masked weights. The CAR method is used to produce pruned models (pruned to 50% and 70%) that have accuracies close to dense models and PCA (certified robustness) that exceeds the dense model. CAR also outperforms several baseline comparison methods.

**Strengths:**

The paper demonstrates relatively strong results on their tasks and models of interest. The authors provide theoretical justification for their method and experimental results, including ablation results.

**Weaknesses:**

- One weakness is the lack of relevant context. There are no results on transformers or other modern architectures so the generalization of results are somewhat limited. As well, recent papers are not referenced (see [1, 2, 3, 4]).
- It seems as if pruning is not a central part of the method. There is no discussion of efficiency, which could be a benefit if the method were shown to work on structured pruning methods. As well, there is a lack of comparison to other state of the art unstructured pruning methods. The authors could compare to the robust pruning literature (eg. see cited papers below).
- Another weakness revolves around the type of robustness being analyzed. There is plenty of work on analyzing various types of out of distribution (non-adversarial) robustness, but the authors chose to apply adversarial robustness techniques to very custom OOD robustness tasks. I would appreciate seeing more discussion of semantic perturbations, justifications for the chosen tasks, and experiments on a larger variety of semantic perturbations. As well, I would be interested to see whether the results hold on other OOD robustness tasks.

[1] https://arxiv.org/pdf/2205.12694
[2] https://arxiv.org/pdf/2306.14306
[3] https://openaccess.thecvf.com/content/CVPR2021W/SAIAD/papers/Vemparala_Adversarial_Robust_Model_Compression_Using_In-Train_Pruning_CVPRW_2021_paper.pdf
[4] https://arxiv.org/pdf/2412.14714

**Questions:**

Questions
1. Why did the authors choose to use adversarial techniques and measures of robustness for a non-adversarial task? Certified robustness is used to provide guarantees of robustness to adversarial attacks, so I would be curious to hear more about the authors’ choice of framing for this method.
2. How is pruning useful in this context? Unstructured pruning cannot be used for efficiency speedups and there is no discussion of ie analyzing the sparse subnetwork found by this method.
3. Does this method offer improvement to other types of robustness tasks? Does the performance of this method hold up under other metrics, ie relative robustness ratio?

---

### Official Review · Reviewer_p97b · 2025-11-01

**Soundness:** 1
**Presentation:** 1
**Contribution:** 2
**Rating:** 0
**Confidence:** 3

**Summary:**

The paper addresses the problem of obtaining compact (compressed) and robust models to semantic perturbations. The proposed method, Compact-Aware semantic Robustness (CAR), is a regularization loss that combines several constraints: stability, soft-hard mask consistency for compression, a margin ratio constraint for robustness, and an L1 penalty. The evaluation on the CelebA-HQ and Flowers102 datasets using ResNet18, GoogleNet, and MobileNetV2 shows improvements over selected baselines. An analysis of the loss landscape is provided to show the flatness of the model obtained with CAR.

**Strengths:**

* Proposing a method for compact and robust models to semantic perturbations is important.

* The paper proposes a method grounded in theory, and the loss-elicitation process is well structured.

**Weaknesses:**

-   Definition scope: “Semantic (natural and adversarial) perturbations” is a broad notion and has been studied in various ways (see a few references below). The authors appear to suggest they are redefining it.

-   Weak evaluation:

    -   Datasets: Semantic perturbations come in different kinds; evaluating only on CelebA-HQ and Flowers-102 is not sufficiently comprehensive. Consider corrupted datasets (CIFAR-10-C, CIFAR-100-C, ImageNet-C).

    -   Models: GoogLeNet is too old to be informative for current evaluation. ResNet-18 and MobileNet-V2 are relatively small in parameter count; a ResNet-50, Wide-ResNet, or a transformer model would be more relevant to current research.

-   Imprecision: Imprecise definitions and equations make it difficult to understand the method and verify the proofs.

-   Some of the proofs may be incorrect, such as the first proof in Appendix A.3.

References

1] Hendrycks, Dan, and Thomas Dietterich. “Benchmarking neural network robustness to common corruptions and perturbations.” ICLR 2019. https://openreview.net/pdf?id=HJz6tiCqYm

[2] Hendrycks, Dan, et al. “Natural adversarial examples.” CVPR 2021. https://arxiv.org/abs/1907.07174

[3] Gowal, Sven, et al. “Achieving robustness in the wild via adversarial mixing with disentangled representations.” CVPR 2021. https://arxiv.org/abs/1912.03192

[4] Ghiasi, Amin, Ali Shafahi, and Tom Goldstein. “Breaking certified defenses: Semantic adversarial examples with spoofed robustness certificates.” ICLR 2020. https://openreview.net/pdf?id=HJxdTxHYvB

**Questions:**

-   Lines 117–118, Definition (Probabilistic Robustness):

    -   The definition starts with a single sample and a single transform T, and S_T is a singleton set—this is inconsistent with the remainder of the definition and with Equation (1).

    -   The proof referenced in Appendix A.2 appears incorrect. In particular, the claim that the right-hand side of the equation is positive is not correct. Could you clarify this point?

-   Line 149, Definition (Model Pruning): Is the mask m_C​ not supposed to be the same size as the parameters?

-   Compression operator: What is the compression operator C, and from which distribution is it sampled in Lines 160–161? It was introduced as an index for compression in the definition of model pruning (Line 149) but is referred to later as a “compression instance” (Line 159) without clear specification.

-   Baselines for semantic perturbations: If you are evaluating semantic perturbations, why is classical adversarial training (Madry et al.) used as a baseline? Are there no methods that directly target robustness to semantic perturbations (e.g., data augmentation)?

---

### Official Review · Reviewer_tLxx · 2025-11-01

**Soundness:** 3
**Presentation:** 2
**Contribution:** 2
**Rating:** 2
**Confidence:** 3

**Summary:**

This paper proposes a post-training compression algorithm, CAR, that performs unstructured pruning and is claimed to be robust to semantic mutations of the input. CAR is shown to provide improvements in semantic robustness on some models compared to baselines.

**Strengths:**

**Strong empirical validation of claims**:
The paper clearly states its claims and research questions, which are answered empirically, demonstrating the advantages of the proposed method.

**Weaknesses:**

**Weak Motivation**:
Although the problem the paper aims to address, compression that is robust to semantic perturbations, is interesting, there appears to be a lack of recent literature addressing this issue in the models explored in this work. One of the key references, HYDRA (2020), is relatively old, and several new algorithms for model compression have been proposed since then.

**Unclear technical content**:
There are several statements that are difficult to follow, and some key terms are not well defined in the paper. Please define all terms when they are first introduced and provide rigorous mathematical statements when required, and provide intuition and interpretation for them afterward.

**Narrow experimental scope**:
The experimental validation is performed only on small vision classification models (e.g., ResNet-18) trained on small datasets (<10,000 images). It is unclear whether the observed improvements over baselines would transfer to larger datasets and models.

**Lack of reproducibility**:
The authors do not provide code or detailed algorithmic implementation details, making it difficult to reproduce the reported results. Please include code or pseudocode for the pruning procedure and specify hyperparameters to aid reproducibility.

**Questions:**

1. Could the authors clarify lines 118-119 and describe what $\mathcal{S}_{T}$ is referring to? It is unclear whether this is a singleton set or a distribution.

2. Could the authors clarify lines 130-133? It is unclear how the measurements between the outputs of a model on an input and its transformed input can imply a statement about the true class label. The proof in Appendix A.2 assumes that the prediction on the true input matches the true label, which should be stated explicitly as an assumption.

3. Could the other provide a formal definition for Semantic Mutation? Lines 139-143 do not present a mathematically rigorous definition.

4. Could the authors clarify what ‘’$C \sim \mathcal{Q}_{\phi}$’’ is, as stated in line 160? Is C sampled from a distribution parameterized by $\phi$? If so, what is this distribution, and how is it parameterized?

5. Could the authors clarify Line 189: ‘’It is isomorphic to Z expression’’? The meaning is unclear, yet it seems to be an important observation used in the algorithm.

6. It is unclear what $C^*$ is referred to. Could the authors define this mask?

7. In lines 243-244, is it not the case that 10% of parameters are set to values greater than or equal to one, given that no clamping is performed?

8. In line 3 of Algorithm 1, it is unclear, based on the three defined loss functions, how the true label $y$ is used. The training loss appears to be independent of the true label.

9. This work focuses on performing unstructured pruning, which, without specialized hardware and software, does not provide any efficiency gains. This raises questions about the practical usefulness of the proposed approach for efficient inference. Can the method be extended to structured pruning?

**Looking forward to the discussion period to clarify these questions and strengthen this work.**

---

### Note · Authors · 2025-11-17

I have read and agree with the venue's withdrawal policy on behalf of myself and my co-authors.